# Platinum Black/Gold Nanoparticles/Polyaniline Modified Electrochemical Microneedle Sensors for Continuous In Vivo Monitoring of pH Value

**DOI:** 10.3390/polym15132796

**Published:** 2023-06-23

**Authors:** Tao Ming, Tingting Lan, Mingxing Yu, Hong Wang, Juan Deng, Deling Kong, Shuang Yang, Zhongyang Shen

**Affiliations:** 1Research Institute of Transplant Medicine, Tianjin First Central Hospital, Nankai University, Tianjin 300190, China; mingtao17@mails.ucas.ac.cn (T.M.); m15342064283@163.com (T.L.);; 2Institute of Biomedical Engineering, Chinese Academy of Medical Sciences & Peking Union Medical College, Tianjin 300192, China

**Keywords:** polyaniline, electrochemical sensor, nanocomposites, continuous in vivo monitoring, pH value

## Abstract

Continuous in vivo monitoring (CIVM) of pH value is essential for personalized medicine, as many diseases are closely related to acid–base imbalances. However, conventional pH meters are limited in their ability to perform CIVM due to excessive blood consumption, large device volume, frequent calibration, and inadequate real-time monitoring. There is thus an urgent need for a portable method for CIVM of pH value. To address this need, we propose a minimally invasive, continuous monitoring solution in the form of an implantable pH microneedle sensor (MNS) in this study. The MNS is based on the integration of an acupuncture needle (AN) and a Ag/AgCl reference electrode. We fabricate the sensor by electrochemically depositing platinum black and gold nanoparticles onto the AN and further modifying it with polyaniline to increase its sensitivity to hydrogen ions. The pH value is obtained by calculating the open circuit voltage between the modified AN and the reference electrode. The resulting MNS demonstrates excellent selectivity and a high nernstian response to pH (−57.4 mV per pH) over a broad range (pH = 4.0 to pH = 9.0). Both in vitro and in vivo experiments have verified the performance of the sensor, showcasing its potential for biomedical research and clinical practice. The MNS provides an alternative to conventional pH meters, offering a less invasive and more convenient way to perform CIVM of pH value. Moreover, this electrochemical implantable sensor based on AN and silver wires provides a simple and sensitive method for continuous in vivo detection of other biomarkers.

## 1. Introduction

Continuous monitoring of physiological parameters in vivo is critical for personalized medicine, and pH value is one of the most important indicators of biological processes [1,2,3]. Imbalances in pH can lead to various diseases, including renal failure, ischemia, and mental illness [4,5]. Therefore, developing a sensitive sensor for continuous in vivo monitoring (CIVM) of pH values is urgently needed to explore the physiological role of pH in acid-base imbalances and monitor the health of patients [6,7,8].

Various methods have been proposed for testing pH values. While these methods are sensitive and accurate, they have some shortcomings when it comes to in vivo monitoring of pH values. For instance, glass electrodes are sensitive and reliable, but their fragile sensing interface and difficulty in miniaturization for in vivo detection can be problematic [9]. Another method involves the use of pH-sensitive dyes incorporated into cells or tissues [10], but the cytotoxic effects of the fluorescence dyes during in vivo imaging may lead to certain inevitable side effects [11].

To address these challenges, electrochemical methods provide a superior approach to realize CIVM of pH values [12]. Electrochemical detection offer the following three characteristics compared to other methods [13,14]. Firstly, the electrochemical sensors allow miniaturization as they do not require a large sensing interface [15,16]. Secondly, electrochemical sensors have no strict requirements on the shape of the sensor, only requiring a complete circuit formation [17]. Finally, the signal of the electrochemical sensor is an electrical signal, which can be transmitted through current wires without restriction [18]. Clearly, these properties make electrochemical sensors highly suitable for CIVM of pH values.

The electrochemical implantable sensors based on acupuncture needles (AN) have gathered increased attention in CIVM of many biomarkers owing to their unique characteristics [19]. For example, AN are extremely thin metal needles, making them strong, flexible, and capable of smooth implantation with minimal trauma. Consequently, AN are considered a safe and effective substrate for a clinical implantable sensor. For example, Tang et al. proposed an acupuncture needle modified with an iron-porphyrin functionalized graphene composite for the real-time monitoring of nitric oxide release [20]. Simolary, Daniela et al. developed a minimally invasive needle-sensor with a high surface area to monitor O_2_ levels in the brain using acupuncture needles [21]. Furthermore, Zhou et al. designed a micro-needle electrode modified with Au nanotubes and carbon nanotubes for real-time monitoring of norepinephrine [22]. AN are simple to prepare, easy to modify and portable, making them attractive sensor substrates for CIVM of biomarkers.

To improve the electrical performance of AN, the tip of the AN can be modified with metal nanoparticles [23,24]. Metal nanomaterials exhibit unique physical and chemical properties that are completely different from those of traditional materials [25,26]. Platinum black (PtB) is widely used for the development of biosensor due to its high surface area, excellent catalytic activity, and remarkable biocompatibility. One of its most common uses is to enhance the electrical properties of the electrode. For instance, Lu et al. modified microelectrode arrays with PtB to decrease impedance and enhance electron transmission [27], and Xiao et al. also adopted PtB to increase the electron transmission capabilities of the electrode [28]. In addition, gold nanoparticles (AuNPs) have received increased attention in the field of nanotechnology due to their unique physical and chemical properties, including high surface area-to-volume ratio, optical properties, and biocompatibility [29]. In particular, the use of AuNPs in biosensors has shown great potential for improving sensitivity and selectivity [30]. AuNPs can be easily synthesized and functionalized with other materials, such as polymers or antibodies, to enhance biosensing performance.

In addition, polyaniline (PANI) has been widely considered as an ideal candidate for pH sensing applications due to its excellent pH-dependent reversible charge transfer [31,32,33]. Moreover, PANI has a low cost and can be easily synthesized, making it an attractive material for pH sensor development. For instance, Rahim Rahimi et al. developed a wearable pH sensor by incorporating PANI as a conductive filler on a polyimide sheet [34], and M. Kaempgen et al. proposed a sensitive pH sensor by depositing PANI on thin film carbon nanotube networks [35]. However, CIVM for pH can only be achieved if the sensor size is small enough. AN is a more suitable substrate. Therefore, in the development of AN-based electrochemical biosensors, PtB can greatly enhance the electrical properties of AN, and AuNPs modification further increases the specific surface area of the electrode, thereby improving the sensitivity of the electrode. As a material sensitive to pH value, polyaniline is modified on the electrode to endow the electrode with the ability to detect the concentration of hydrogen ions.

In this paper, we propose a pH microneedle sensor based on the integration of an acupuncture needle modified with PANI/AuNPs/Ptb and a Ag/AgCl reference electrode. The use of an acupuncture needle as the base structure provides advantages such as small size and minimal tissue damage upon penetration. The modification of the acupuncture needle with AuNPs/Ptb improves the conductivity of the electrode and increases its specific surface area, enabling highly sensitive detection of hydrogen ions. The electrical state of the electrode, deposited by polyaniline, changes with the concentration of hydrogen ion. The Ag/AgCl reference electrode provides a stable and reliable reference potential for continuous monitoring of pH values in vivo. The proposed sensor demonstrates a high nernstian response to pH (−57.4 mV/pH) over a broad range (pH = 4.0 to pH = 9.0). We believe that this novel electrochemical sensor can provide a sensitive and reliable method for clinical CIVM of pH value.

## 2. Material and Methods

### 2.1. Reagents

Acupuncture needles (Diameter 0.3 mm, length 40 mm) were obtained from Tianjin Jiangtian Chemical Technology Co. Ltd. (Tianjin, China), Hydrogen tetrachloroaurate (III) tetrahydrate (HAuCl_4_·4H_2_O), chloroplatinic acid (H_2_PtCl_6_), lead acetate ((CH_3_COO)_2_Pb), magnesium chloride (MgCl_2_) and aniline were obtained from Macklin Biochemical Technology Co., Ltd. (Shanghai, China). Sodium chloride (NaCl), potassium chloride (KCl), and anhydrous ethanol were obtained from Shanghai Aladdin Biochemical Technology Co., Ltd. (Shanghai, China). Sulfuric acid (H_2_SO_4_) was purchased from Tianjin Bohai Chemical Reagent Co., Ltd. (Tianjin, China). All other chemical materials were of analytical reagent grade. Standard serum was obtained from Shanghai Epizyme Biomedical Technology Co., Ltd. (Shanghai, China). Experimental mice were obtained from Beijing Vital River Laboratory Animal Technology Co., Ltd. (Beijing, China).

### 2.2. Apparatus

Cyclic voltammetry (CV), Electrochemical Impedance Spectroscopy (EIS), Amperometric i-t Curve (IT) and Open Circuit Voltage-Time (OPCT) were implemented on a CHI660E electrochemical workstation (Chenhua Co., Ltd., Shanghai, China). A pH meter (Beijing JD Industrial Products Trading Co., Ltd., Beijing, China) was used for adjusting the pH values in solution. An S-3500 scanning electron microscope (S-3500, Hitachi, Tokyo, Japan) was used to record the SEM images. An ultrasonic generator was used to obtain homogeneous solutions (Labgic, Beijing, China).

### 2.3. Fabrication of the Microneedle Sensors

#### 2.3.1. Fabrication of the AN Working Electrode

The sensing window of the AN is located at the tip and has a size of 6 mm. At the other end, there is a 1.0 cm conductive part for connection with the electrochemical instrument. The remaining portion of the AN is insulated with epoxy resin. The AN is expected to be implanted into the upper arm. For the preparation of the AN, it was first sonicated in anhydrous ethanol and deionized water for 30 min to remove any inorganic compounds and organic impurities from the electrode surface. Next, 3 wt% chloroplatinic acid solution was mixed with 0.25 wt% lead acetate solution to obtain platinum plating solution. Then, Ptb was deposited on the surface of the actual sensing window by electrochemical polymerization (IT, −2.5 V, 100 s). This modified AN is denoted as Ptb/AN for subsequent modification.

Subsequently, AuNPs were deposited on the Ptb/AN by electrochemical polymerization (CV, −0.2 V to 0.5 V, 50 mV/s, 35 cycles). The resulting electrode is denoted as AuNPs/Ptb/AN. Finally, H_2_SO_4_ (1 mol/L) was adapted as the supporting electrolyte, and PANI was electrochemically deposited on the AuNPs/Ptb/AN in aniline solution (0.1 mol/L) (CV, −0.2 V to 0.2 V, 5 mV/s, 4 cycles). The pH sensitive working electrode of PANI/AuNPs/Ptb/AN was fabricated for further use.

#### 2.3.2. Fabrication of the Ag/AgCl Reference Electrode

The silver wire with a diameter of 0.5 mm was cut into 40 mm lengths to create the Ag/AgCl reference electrode. To prepare the electrode, the silver wire was first sonicated in anhydrous ethanol and deionized water for 30 min to remove inorganic compounds and organic impurities from its surface. Next, AgCl was electrochemically deposited on the surface of the silver wire in a saturated potassium chloride solution (IT, 0.5 V, 2 s), and the Ag/AgCl reference electrode was fabricated.

Finally, the PANI/AuNPs/Ptb/AN and the Ag/AgCl reference electrode were bonded together using a hot melt gun. The detailed schematic is shown in Figure 1.

## 3. Results and Discussion

### 3.1. Characterization of the Modified Working Electrode

Appendix A shows the physical image of the bare AN (Appendix A), Ptb/AN (Appendix A), AuNPs/Ptb/AN (Appendix A) and PANI/AuNPs/Ptb/AN (Appendix A). These images clearly illustrate the changes that occur after each layer-by-layer modification. Additionally, the elemental changes on the electrode surface are confirmed by the results of energy-dispersive X-ray spectroscopy (EDS). The corresponding scanning electron microscope (SEM) image of those four electrode is shown in Figure 1. It can be clearly seen that with the modification of nanomaterials, nanostructures have formed on the electrode surface, and the specific surface area of the electrode has significantly increased.

The CV response of PANI/AuNPs/Ptb/AN in 5 mM K_3_Fe(CN)_6_ solution containing 0.1 M KCl at different scan rates (20–200 mV/s) were performed to evaluate its electrochemical behavior [36]. Figure 2A,B depict the results obtained from the CV analysis. It is observed that both the anodic and cathodic peak currents displayed a linear relationship with the square root of the scan rate (R^2^ = 0.996 and 0.997, respectively). It indicated a diffusion-controlled behavior. Additionally, by examining the peak separation in the CV curves shown in Figure 2A, it can be inferred that the reaction is quasi-reversible.

### 3.2. Electrochemical Performance of the Proposed MNS

Appendix A illustrate the schematic of the measurement setup that will be discussed below. Figure 3 presents the gradually changing electrical properties of the electrode. The CV response of bare AN, Ptb/AN and AuNPs/Ptb/AN in 5 mM K_3_Fe(CN)_6_ solution containing 0.1 M KCl were shown in Figure 3A, and the voltage difference between the open circuit voltage (pH = 6 and pH = 8) detected by bare PANI/AN, PANI/Ptb/AN and PANI/AuNPs/Ptb/AN were shown in Figure 3B. Obviously, with the layer-by-layer modification of the electrode, the electrical performance of the electrode significantly increases, and the sensitivity of the electrode to hydrogen ions increases accordingly. As shown in Figure 3C, by integrating PANI/AuNPs/Ptb/AN and Ag/AgCl reference, the as-prepared MNS shows significant pH value detection capability.

### 3.3. Analytical Performance of the Proposed MNS

The analytical performance of the MNS was assessed by measuring standard pH buffer solutions. The OPCT responses of the MNS to a series of pH value and the corresponding calibration curves are shown in Figure 4. Figure 4A illustrate that higher pH value corresponded to lower concentration of hydrogen ions, resulting in a lower open circuit voltage. The calibration plots in Figure 4B demonstrate a linear relationship between the open circuit voltage (Voc) and the pH value (X), described by the equation Voc (Y) = 0.6877 − 0.0574X. The correlation coefficient is 0.999, and the error bar indicates standard deviation. The resulting MNS demonstrated excellent selectivity and a high nernstian response to pH (−57.4 mV per pH) over a broad range (pH = 4.0–9.0). These results illustrate that the proposed MNS can be used to sensitively and directly detect pH value.

### 3.4. Real-Time Monitoring toward pH

The proposed MNS’s ability of real-time monitoring of pH value was evaluated by measuring standard pH buffer solutions and standard serum. Appendix A illustrates the real-time OPCT responses in various pH buffer solutions. It is observed that the potential gradually increases with a decrease in pH in phosphate-buffered saline buffer solutions from pH 9.0 to pH 4.0. Conversely, when the pH is changed from 4.0 to 9.0, the corresponding potential gradually decreased with an increase in pH. Minimal drift is observed at the same pH values within a cycle, indicating the reliable performance of the MNS for real-time pH monitoring.

In addition, the pH value of the blood was adjusted by adding 40 μL NaH_2_PO_4_ solution (0.5 mg mL^−1^) and 40 μL of Na_2_CO_3_ (0.5 mg mL^−1^) solution. Similarly, as the pH is increased by adding Na_2_CO_3_ solution, a decrease in OPCT is observed in serum. Conversely, as the pH decreases with the addition of NaH2PO4 solution, an increase in OPCT response is observed (Figure 5). The pH changes resulting from the addition of NaH_2_PO_4_ and Na_2_CO_3_ are estimated to be −0.47 and 0.61, respectively. The typical response time was estimated to be approximately 420 ms, indicating the fast response of the MNS. These results confirm that the proposed MNS enables real-time monitoring of pH values.

### 3.5. Continuous In Vivo Monitoring of pH Value In Vivo

To further assess the ability of continuous in-vivo monitoring of pH value, measurements were conducted in the abdominal main vein of rats by the proposed MNS (Figure 6). The proposed MNS was implanted into the abdominal main vein, and the modified working electrode, along with the reference electrode, were connected to the electrochemical workstation via a green conductive wire and a yellow conductive wire. It is crucial to ensure that the two electrodes do not bend or come into contact with each other during the implantation process.

The pH value of the blood in the abdominal main vein was adjusted by injecting 40 μL of NaH_2_PO_4_ solution (0.5 mg mL^−1^) and Na_2_CO_3_ solution (0.5 mg mL^−1^) using a syringe pump. As shown in Figure 6B, upon injection of Na_2_CO_3_ and NaH_2_PO_4_ solutions, an immediate decrease and increase in the corresponding open circuit voltage were observed, respectively. These results demonstrate that the prepared MNS can be effectively utilized for continuous in vivo monitoring of pH changes in the blood during physiological processes.

The proposed sensor has the following limitations in its application. First, because the sensor is composed of needle and reference electrodes, it will cause greater trauma than expected when implanted. Secondly, the detection also requires the proposed sensor to be connected to an electrochemical workstation, so it is not convenient enough.

### 3.6. Selectivity, Repeatability and Stability

Selectivity is a crucial property for biosensors, especially for a sensor that is about to be applied in blood. The selectivity of the proposed MNS was evaluated by investigating by the interference of Na^+^, K^+^ and Mg^+^ with H^+^. Since these ions are commonly present in blood, it is crucial to assess the sensor’s selectivity in the presence of these interferences. The proposed MNS was exposed to 2 mM NaCl, 2 mM KCl, 2 mM MgCl_2_ and 100 μM H^+^ solutions, and the open circuit voltage changes were monitored through OPCT. As shown in Figure 7A, the addition of NaCl, KCl and MgCl_2_ resulted in minimal changes in the open circuit voltage. However, when the H^+^ was added at a concentration one-twentieth of the aforementioned solutions, a significant increase in the open circuit voltage was observed. Figure 7B further illustrates the impact of different ions on the open circuit voltage measured by the proposed MNS. The changes in open circuit voltage caused by Na^+^, K^+^ and Mg^+^ were only 2.75%, 0.75%, and 2.5% of that caused by H^+^, indicating excellent selectivity of the proposed MNS.

Repeatability, another crucial property for biosensors, was also evaluated for the proposed MNS. Three separate measurements were conducted using pH buffer solutions with a pH value of 6, 7 and 8. As shown in Figure 7C, in the pH buffer solutions (pH = 6), the specific values of the open circuit voltage were 359.4, 354.5, 355.9 mV, respectively. In the pH buffer solutions (pH = 7), the specific values of the open circuit voltage were 284.6, 286.3, 285 mV, respectively. In the pH buffer solutions (pH = 8), the specific values of the open circuit voltage were 228.6, 216.2, 229 mV, respectively. As shown in Figure 7D, the standard deviation from these measurements was 2.52, 0.89 and 7.28 mV, and the coefficient of variation from these measurements was 0.71%, 0.31%, 0.59, and 3.24%, respectively. These results demonstrate the excellent repeatability of the proposed MNS.

The stability of the proposed MNS, particularly in blood, was assessed by measuring its changes after being stored in standard serum for seven days. The CV response of the proposed MNS and the MNS after storing in standard serum for seven days in 5 mM K_3_Fe(CN)_6_ solution containing 0.1 M KCl were recorded and compared, as shown in Appendix A. After 7 days of storage, the anodic and cathodic peak currents decrease by 15.18% and 8.37%, respectively. Appendix A demonstrates the change in the sensor’s detection capability, where the voltage difference between pH 6 and pH 8 decreased by 15.99% after seven days of storage. These results indicate that the proposed MNS exhibits good stability even after being stored in standard serum for an extended period.

### 3.7. Comparation of Different Sensors

There are many previous studies that is aimed at the development of pH sensing devices. Table 1 compares our MNS and several other electrochemical pH biosensors in terms of their sensing substrate, the adapted nanomaterials, their sensitivity, and the detection range. The proposed MNS demonstrates excellent performance.

## 4. Conclusions

In this work, we proposed an electrochemical pH microneedle sensor based on the integration of an acupuncture needle modified with PANI/AuNPs/Ptb and a Ag/AgCl reference electrode. Compared with many previously described pH sensors, this work features several innovations. (1) We have streamlined the process of preparing the microneedle sensor. Only simple treatment of acupuncture needles and silver wires is required, followed by a few steps of electrodeposition for modification. The acupuncture needles and silver wires are then bonded by melting glue to realize the preparation of the microneedle sensor. These facile preparation processes make the proposed sensor suitable for large-scale production and inexpensive to fabricate. (2) The extremely fine structure of acupuncture needles and silver wires reduces the trauma caused by implantation, and the signal acquisition does not require any further in vivo manipulations. These characteristics make the proposed sensor ideal for implantation for continuous in vivo monitoring of pH values. (3) PANI/AuNPs/Ptb/AN was successfully fabricated. The proposed MNS is selective, sensitive, and biocompatible, and is a portable, accurate, reliable and inexpensive tool for pH monitoring in vivo. Additionally, this integrated electrochemical implantable sensor, based on AN and silver wires, provides a simple and sensitive method for continuous detection for other biomarkers.

However, the proposed sensor does have some limitations. For example, the working electrode and the reference electrode need to be implanted in the body at the same time, which increases the trauma during implantation. In addition, the reading of the signal depends on an electrochemical workstation, which makes the detection of the pH value not yet portable. To address these limitations, we plan to focus our future work on how to prepare the working and the reference electrode on the same acupuncture needle to minimize trauma during implantation. We will also develop a portable electrochemical reading equipment to remove the shackles of the electrochemical workstations, and make pH detection truly portable.

Overall, the proposed pH microneedle sensor shows great potential as a portable and inexpensive biosensor capable of CIVM of pH values. This simple and easy to fabricate novel implantable sensor will provide a reference for the development of in vivo real-time monitoring sensors for various disease biomarkers.

## Data Availability

Data is contained within the article.

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
