# Peer review of "Platinum Black/Gold Nanoparticles/Polyaniline Modified Electrochemical Microneedle Sensors for Continuous In Vivo Monitoring of pH Value"

_polymers, 2023, doi:10.3390/polym15132796_

Round 1
Reviewer 1 Report
This paper discusses the development of an electrochemical microneedle sensor modified with platinum black/gold/polyaniline nanoparticles for the continuous in vivo monitoring of pH levels.
The physical and electrochemical properties of the developed layers were discussed in detail, and sensor properties such as sensitivity, repeatability, and stability were also discussed. The performance of the sensor was verified using in vitro and in vivo experiments.
In general, the paper is good and the topic of the study is interesting. I would like to accept this paper for publication. However, some issues need to be clarified:
1. It is not clear why these three materials need to be placed on the AN surface. Thus, it is necessary to explain the specific role of each layer.
2. It is not clear how and in which part of the body the needle was implanted. It is also not clear how long is the needle. If it is implanted under the skin, it should be ensured that it is not too long to avoid it reaching the nervous system.
2. Why a silver wire with AgCl coating is chosen as a reference electrode?. How about some other material, such as Gold, Pt, etc that have better electrical conductivity and high chemical stability.
3. Figure 1 (a,b,c,d) show only the physical structure of the nanocomposite. An EDAX/XRD analysis is necessary to confirm the presence of nanoparticles and the structure of the composite.
4. It is necessary to show the schematic of the measurement setup discussed in subsections 3.2 to 3.5.
5. it has been mentioned that the coefficient of variation of the measurement varies from 0.71% to 3.24% which shows excellent repeatability. What is the standards ? and up to how many % it is still acceptable?.
Reviewer 2 Report
The authors demonstrate a nanocomposite functionalized electrode for in-vivo pH sensing. This topic is very interesting as there is no current a reliable solution to do continuous pH measurements inside the animal or human body. The manuscript could be suitable for publication after minor modifications.
1- What type of CV is being generated by the electrode. Is it reversible or quasi-reversible? This would give an electrochemical insight of how efficient the pH electrode fabrication is. Please calculate using peak separation from CVs in Fig 2A.
2- Please report the result from section 3.5 in the main manuscript and not supporting info as this is a main finding of the paper. Please discuss limitations.3
Please revise language as it is broken in many sentences such as sentence in section 3.5 where the authors state “It is worth noting that do not allow the two electrodes”. Others exist as well.
